# Characterization of the bacterial fecal microbiota composition of pigs preceding the clinical signs of swine dysentery

Jéssica A. Barbosa[1,2], Juan C. P. Aguirre[2], Roman Nosach[2], John C. S. Harding[2], Vinícius S. Cantarelli[1], Matheus de O. Costa[2,3]*

1 Animal Science Department, Federal University of Lavras, Lavras, Minas Gerais, Brazil, 2 Department of Large Animal Clinical Sciences, Western College of Veterinary Medicine, University of Saskatchewan, Saskatoon, SK, Canada, 3 Department of Population Health Sciences, Faculty of Veterinary Medicine, Utrecht University. Utrecht, the Netherlands

* matheus.costa@usask.ca

**Data Availability Statement:** Minimal anonymized dataset is publicly available online from the OSF

## Abstract

Swine dysentery (SD) is a worldwide production-limiting disease of growing-finishing pigs in commercial farms. The importance of the large intestinal microbiota in the swine dysentery pathogenesis has been established, but not well characterized. The objective of this study was to characterize the fecal bacterial microbiota of pigs immediately prior to developing clinical signs of swine dysentery. A total of 60 fecal samples were collected from 15 pigs with SD. Sampling times included a time point prior to SD (d0, n=15), 2 days before mucohaemorrhagic diarrhea was observed (d-2SD, n=15), 1 day before mucohaemorrhagic diarrhea was observed (d-1SD, n=15), and the day when pigs developed mucohemorragic diarrhea (MHD, n=15). Sequencing of *cpn*60 amplicons was used to profile the microbiome, and analyses were performed on QIIME2. Increased Chao1 index in d-1SD and MHD samples when compared to the d0 was the only change observed in alpha diversity. No differences between sampling times on beta diversity (Bray-Curtis dissimilarity) were found. Although a small sample size was investigated, differential abundance analysis revealed that *Alistipes dispar* and *Parabacteroides gordonii* were increased in MHD fecal samples when compared to d-2SD and d-1SD. It is suggested that these taxa may play a role in the pathogenesis of SD, which is known to require the presence of *Brachyspira* spp. and an anaerobe for severe disease development.

## Background

Swine dysentery (SD) is a production limiting enteric disease affecting grower-finisher pigs worldwide. The anaerobic tolerant spirochete *Brachyspira hyodysenteriae* was the etiological agent of SD [1], which is characterized by mucohaemorrhagic diarrhea and fibrinonecrotic colitis [1, 2]. Recently, *B. suanatina* [3] and *B. hampsonii* [4] were also recognized as agents of SD.

A complex interaction between the large intestine microbiota and *B. hyodysenteriae* has been demonstrated. Studies based on culture have shown that *B. hyodysenteriae* requires the

repository (https://doi.org/10.17605/OSF.IO/8TSX6).

**Funding:** MOC - Natural Sciences and Engineering Research Council of Canada - NSERC Discovery - RGPIN-2020-06353. https://www.nserc-crsng.gc.ca/index_eng.asp The funders had no role in study design, data collection and analysis, decision to publish, or preparation of the manuscript.

**Competing interests:** The authors have declared that no competing interests exist.

presence of other bacteria for severe SD expression [5–9]. Several studies provided evidence that gnotobiotic pigs inoculated with *B. hyodysenteriae* alone did not develop mucohaemorrhagic diarrhea [5, 10]. Colitis and mucohaemorrhagic diarrhea was reported in gnotobiotic pigs co-inoculated with *B. hyodysenteriae* and *Fusobacterium necrophorum* or *Bacteroides vulgatus* [5]. Mucoid diarrhea was found in pigs co-inoculated with *B. hyodysenteriae*, *F. necrophorum*, *B. vulgatus*, a *Clostridium* species, or *Listeria denitrificans*, but not when the spirochete was the only inoculum [9]. Recently, high-throughput sequencing of microbial barcode genes, such as 16S rRNA or *cpn*60, has been applied to study SD. Colonic contents and mucosal scrapings from pigs inoculated with *B. hyodysenteriae* or *B. hampsonii* and clinical signs of SD had lower species richness than their uninoculated counterparts [11]. Lower Bacteroidetes:Firmicutes ratio was linked to mucohaemorrhagic diarrhea following inoculation of pigs with *B. hampsonii* compared with sham-inoculated control or inoculated pigs that did not develop clinical disease [12]. *Campylobacter* spp., *Mogibacterium* spp., *Brachyspira* spp., and *Desulfovibrio* spp. were found in higher numbers in mucosal scrapings of pigs that developed SD, whereas *Bifidobacterium* spp. and *Lactobacillus* spp. were significantly more abundant in pigs without SD [11].

It has also been suggested that diet may influence the colonic microbiome and the incidence of SD. However, results have been contradictory [13–16]. Highly fermentable soluble fiber has been associated with a protective effect against SD [17–19]. These diets promoted the growth of lactic acid and butyric acid-producing bacteria, such as *Bifidobacterium* spp., *Megasphaera* spp., and *Faecalibacterium* spp. [16, 20, 21]. Conversely, poorly fermentable insoluble fiber has been linked to increased odds of pigs developing SD [22]. This observation was combined with augmented loads of anaerobes such as *Shuttleworthia* spp., *Ruminococcus torques*, and *Mogibacterium* spp., which may play a synergistic role with *B. hyodysenteriae* in inducing SD [21].

It is well established that the colonic microbial community changes following the development of SD, in comparison to pigs in a clinically healthy state [9, 11, 12, 16]. However, little is known regarding the changes preceding the clinical signs of SD. We hypothesize that the fecal microbiome is disturbed 24 to 48 hours before clinical SD is observed. Thus, the goal of this study was to characterize the bacterial fecal microbiome composition of pigs immediately prior to developing clinical signs of swine dysentery, during the disease incubation period.

## Methods

### Animal trial and samples

Barrows (n=15), age 9 to 10 weeks-old, were obtained from a PRRSV, *Mycoplasma hyopneumoniae* negative, high-health herd farm with no gastrointestinal clinical signs and no history or previous laboratory diagnosis of SD. Animals were housed and allowed to acclimate in a BSL-2 animal care facility for 7 days prior to inoculation. An unmedicated, commercial starter diet fed *ad libitum* was used [23] (S1 Table). Fecal samples were collected from contact pigs that were exposed to diseased seeder pigs experimentally inoculated intragastrically thrice over 72 hours with 100 mL of a 24h broth culture of *B. hyodysenteriae* strain G44 (total dose 3.72 x $10^{11}$ genome equivalents/mL). Contact pigs were sampled one day after contact with seeder pigs (d0, n=15); 2 days before mucohaemorrhagic diarrhea was observed (d-2SD, n=15); 1 day before mucohaemorrhagic diarrhea was observed (d-1SD, n=15); and the day when pigs developed mucohemorragic diarrhea (MHD, n=15), totaling 60 samples. A summary of the samples used from this trial is shown on S1 Table. All fecal samples were collected by digital stimulation and stored at -80°C until processing for analysis. The development of swine dysentery was confirmed by associating clinical signs, a positive fecal *B. hyodysenteriae* culture and gross necropsy lesions.

## DNA extraction, *cpn*60 amplification and sequencing

For each sample, total DNA was extracted from 200 mg of feces using a commercial kit (Mag-Max DNA Ultra v2.0; Applied Biosystems, Thermo Fisher Scientific, Waltham, MA, USA) on a KingFisher Flex platform (Thermo Fisher Scientific, Waltham, MA, USA). Amplification and indexing of the *cpn*60 universal target barcode region were performed as previously described [24]. Briefly, the *cpn*60 gene was amplified using a primer mix comprised of 100 μM from each of the following primers: M279 Forward (5' – `GAIIIIGCIGGIGAYGGIACIACI AC` – 3'), M280 Reverse (5' – `YKIYKITCICCRAAICCIGGIGC`– 3'), M1612 Forward (5' – `G AIIIIGCIGGYGACGGYACSACSAC`– 3'), and M1613 Reverse (5' − `CGRCGRTCRCCGAAG CCSGGIGCCTT`– 3'). Primers were mixed in a 1:3 molar ratio of M279:M280 (3 μL each), and M1612:M1613 (9 μL each) and diluted in 276 μL of ultrapure water for a total volume of 300 μL. PCR reactions had a total reaction volume of 50 μL, for 2 μL of DNA template. The master mix was prepared using 38.1 μL ultrapure water, 5 μL of 10x PCR buffer, 2.5 μL of MgCI2 (50mM), 0.4 μL of Platinum Taq Polymerase (Invitrogen, Thermo Fisher Scientific, USA), 1 μL of dNTP mix (10 mM; Invitrogen, Thermo Fisher Scientific, USA) and 1 μL of the primer cocktail described above. Reactions were incubated at 95˚C for initial denaturation for 5 minutes, followed by 40 cycles of denaturation at 95˚C for 30 seconds, annealing at 60˚C for 30 seconds, extension at 72˚C for 30 seconds, and a final extension at 72˚C for 2 minutes. *Cpn*60 amplicons were purified using NucleoMag NGS beads (Macherey-Nagel Inc., Germany). Indexing PCR for library preparation was performed using a Nextera XT primers library preparation kit (Illumina Inc., San Diego, CA, USA) according to the manu-facturer's protocol. Indexed amplicons were size-selected using NucleoMag NGS beads. Indexed amplicons were quantified, normalized, and diluted to 10pM libraries containing 5% PhiX DNA, Sequencing was carried out on an Illumina MiSeq (Illumina Inc., San Diego, CA, USA) platform using a 500-cycle reagent kit v2 (401 R1, 101 R2, Illumina Inc., San Diego, CA, USA).

## Sequencing data analysis

Following sequencing, on-rig quality control procedures were executed. Raw data was pro-cessed by initially removing sequencing and amplification primers using Cutadapt [25], and low quality or short length and technical sequences were trimmed with Trimmomatic [26]. Fil-tered sequence reads were imported to Quantitative Insights Into Microbial Ecology 2 (QIIME2) [27], and variant calling was carried out using DADA2, truncating at 150 bp from the 5′ end [28]. Reads were mapped to the nonredundant version of cpnDB (cpnDB_nr) using watered BLAST [29]. Downstream analysis used only amplicon sequence variants (ASV) with >55% sequence similarity to a cpnDB_nr match. An feature table was generated and analyzed using a web-based platform for high-throughput sequencing data statistical analyses [30]. For further statistical analysis and visualization, the ASV table with taxa and metadata file were uploaded to the MicrobiomeAnalyst tool (Xia Lab, McGill University, Quebec, Canada; avail-able at: https://www.microbiomeanalyst.ca) [31]. At the data filtering step, a low count filter was used to filter all ASV features with <4 counts in at least 20% prevalence, and 10% mini-mum variance among samples, leaving 169 ASV [30]. Shapiro-Wilk test was used to evaluate the normality of data. The alpha diversity indices (Chao1 and Shannon's index) were calcu-lated on raw data and comparisons were performed using ANOVA followed by post-hoc Tukey test. Beta-diversity differences between groups were analyzed by permutational multi-variate analysis of variance (PERMANOVA) using Bray-Curtis dissimilarity index. Principal coordinate analysis (PCoA) was used to visualize the beta diversity data (MicrobiomeAnalyst). Differential abundance analysis at the phylum and ASV levels between sampling days was

performed using DESeq2 [32], and the adjusted *p*-value<0.05 was used to report the significance (R version 4.2.1, RStudio, Boston, MA, USA, Love et al., 2014).

## Results

Following quality control steps, sequencing resulted in 2,639,963 high-quality reads (average 44,745 per sample, ranging from 4,755 to 86,947). One sample, 387_MHD (rep 8), was removed from the analysis due to an extremely low number of reads generated (159). A total of 589 ASV were detected, 420 had ≥ 2 reads total and were kept for downstream analyses.

### Fecal microbial community composition

Proportional taxa abundance data at the phylum and family level are shown in Fig 1A and 1B, respectively. The top 3 most abundant phyla were firmicutes, terrabacteria-group and bacteroidetes. The top 3 most abundant family were *Bacteroidales*, *Clostridiales* and *Ruminococcaceae*.

No significant differences in alpha diversity indexes (Chao1, *P* = 0.056, Shannon's index, *P* = 0.248) were observed between days at the phylum level (Fig 2A, 2B). At the genus level, Chao1 index was increased at d-1SD (*P* = 0.042) and MHD (*P* = 0.001) when compared to d0 samples (Fig 2C), and no significant differences in Shannon's index were observed among days (Fig 2D; *P* =0.270). No changes in beta diversity were observed at the phylum level. At the genus level, four samples clustered separately from the others significantly affecting the data distribution, but an even distribution of high and low ranks within and between days was observed as evidenced by the small $R^2$ value identified ($R^2$ = 0.091684, *P* = 0.003, Fig 3).

### Differential abundance analysis

Reads associated with actinobacteria and spirochetes significantly differed between days (Fig 4A and 4B). ASVs with more than 10.000 total read counts and differentially abundant between d-2SD vs. MHD, d-1SD vs. MHD and d0 vs. all other days are presented in Tables 1 and 2 and S2 Table, respectively.

## Discussion

This study characterized the fecal microbiota of pigs on the two days immediately prior to the development of swine dysentery clinical signs. Alpha and beta diversity were not significantly on the 2 days prior the observation of mucohaemorrhagic diarrhea. However, differential abundance analysis revealed ASVs significantly affected prior to the observation of clinical SD. This work is an intermediary step towards the complete understanding of SD pathogenesis, and the role of the microbiome in this mechanism.

A high abundance of *Eubacterium brachy* and a low abundance of *Parabacteroides gordonii* was observed in d-2SD, when compared to MHD samples. The *Eubacterium* genera is known to be present in the healthy mammalian intestinal microbiota, as demonstrated in mice, humans and pigs [33–35]. In pigs, *P. gordonii* was found in feces of healthy animals and suggested to have an effect on growth performance [36–38]. Oral administration of *P. distasonis* membrane fractions relieved intestinal inflammation in mice with acute and chronic colitis induced by dextran sodium sulfate [39]. In the same study, the authors observed a reduction in TNF-a production by macrophages *in vitro* [39]. Relative abundance of *P. distasonis* was found inversely associated with colonic production of IL-1β [40]. However, members of this fairly new genus (established in 2006) have been associated with a dichotomous role in intestinal and systemic disease [41]. It remains unclear how *P. gordonii* may have contributed to the

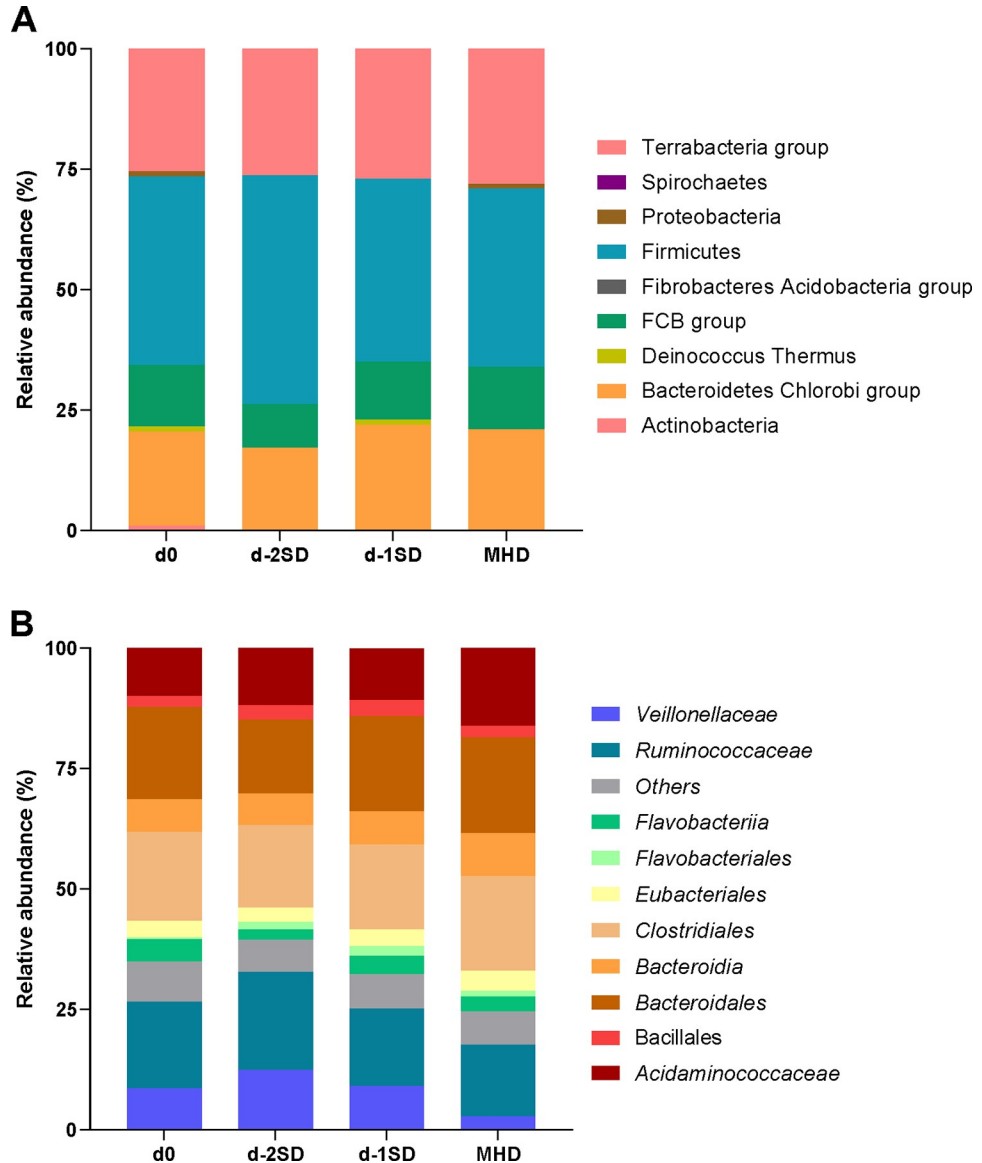

**Fig 1. Stacked bar charts representing proportional abundance of bacterial taxa one day after contact with seeder pigs (d0, n=15); 2 days pre-SD (d-2SD, n=15); 1 day pre-SD (d-1SD, n=15); and the day mucohaemorrhagic diarrhea was observed for the first time (MHD, n =14).** (A) Depicts data at the phylum level, and (B) at the family level.

pathogenesis of SD, and further studies are suggested to try and elucidate its potential relationships with *Brachyspira* spp. *Eubacterium* spp. is suggested to benefit the host largely due to its production of butyrate [42–44]. *Eubacterium brachy*, a Gram-positive strict anaerobe, was frequently isolated from patients with periodontitis and pleuropulmonary infection [45, 46]. Mouse models of colitis using dextran sodium sulfate revealed that *E. limosum* and its metabolites were associated with reduced clinical scores through increased butyrate levels. In T84 colonocyte cells, this effect is mediated by reduced IL-6 and TLR4 expression [47]. Additionally, lower abundance of *Eubacterium* spp. has been reported in patients with ulcerative colitis or Crohn's disease, when compared to healthy patients [48, 49]. Previous work showing the association between a higher abundance of *Eubacterium* spp. and attenuated colitis

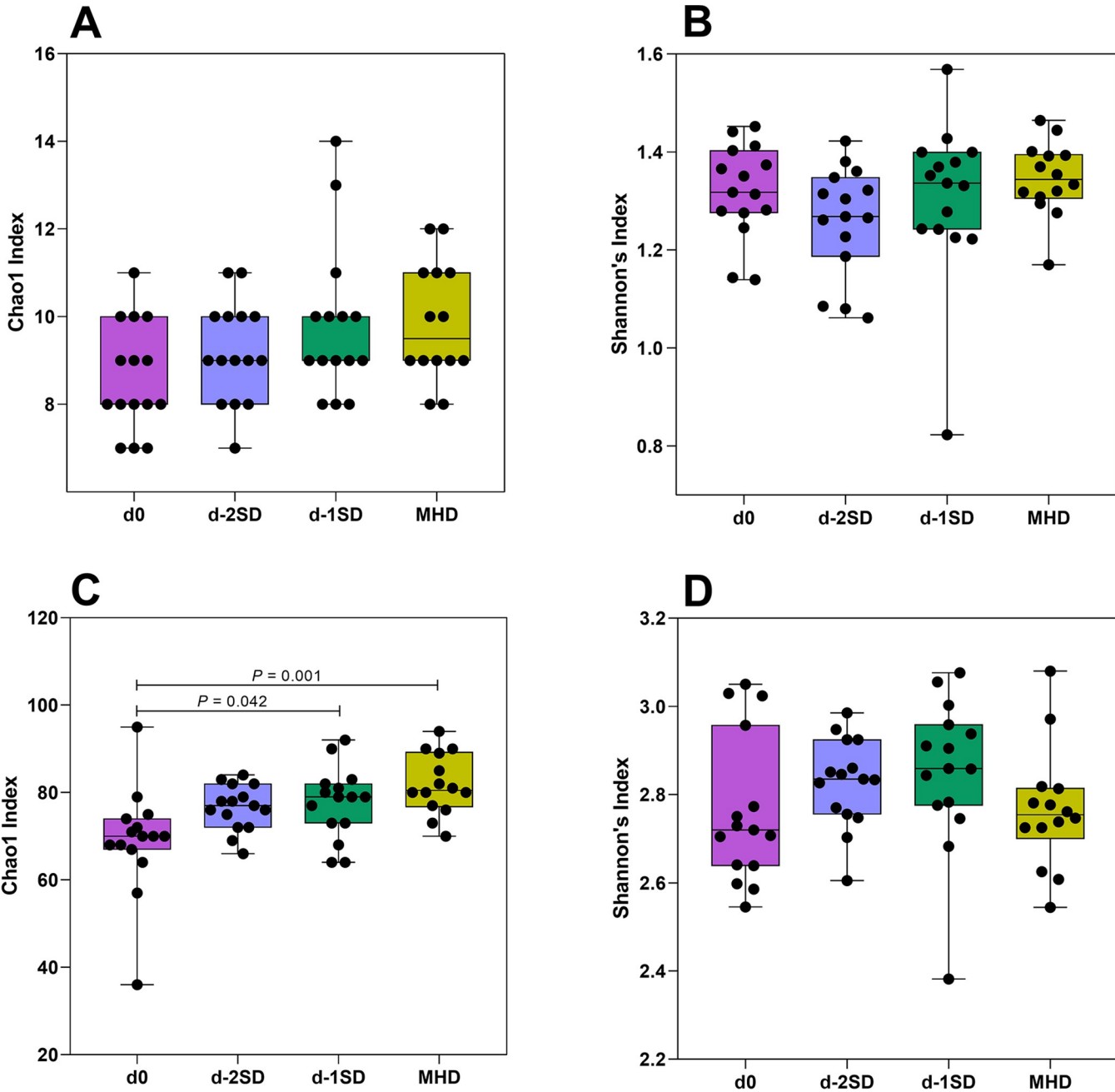

**Fig 2. Alpha-diversity metrics from fecal samples collected after contact with seeder pigs (d0, n=15); 2 days pre-SD (d-2SD, n=15); 1 day pre-SD (d-1SD, n=15); and the day mucohaemorrhagic diarrhea was observed for the first time (MHD, n =14).** A: Chao1 richness at the phylum level; B: Shannon's diversity index at the phylum level; 2C: Chao1 richness index at the genus level; D: Shannon's diversity index at the genus level. Boxes shows interquartile ranges, whiskers depict the minimum and maximum values.

corroborates our findings, as this genera was found depleted in MHD samples. In contrast, *Parabacteroides* spp. is found in low amounts in the human gut microbiota [50]. In pigs, it is suggested to be part of the intestinal microbiota of healthy animals [36–38]. Humans with IBD have decreased abundance of *Parabacteroides*, when compared to healthy patients [51, 52]. Bacteria of this genera were found to produce short chain fatty acids (SCFA) [53], regulate

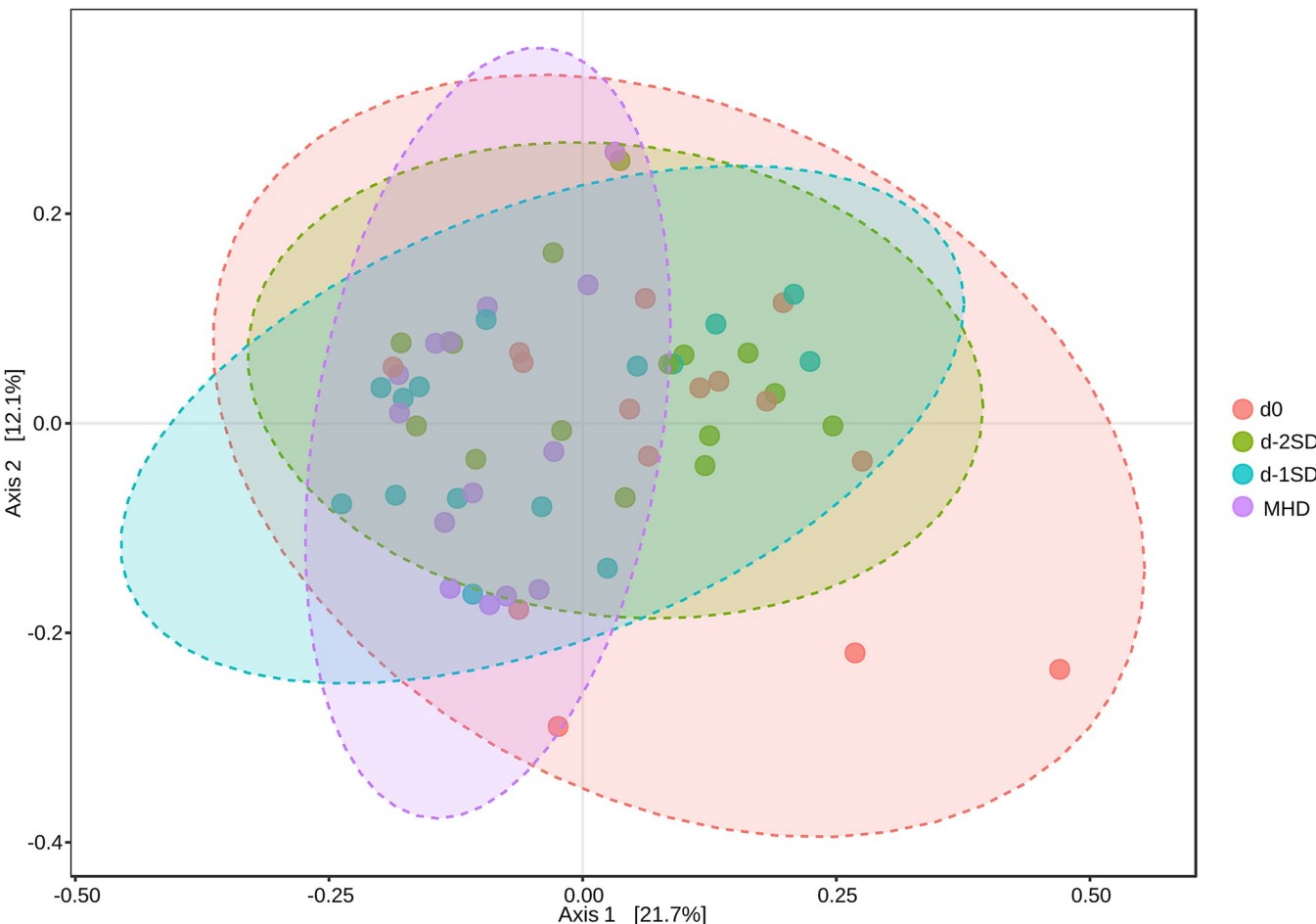

**Fig 3. Beta diversity (Bray-Curtis dissimilarity data) at genus level based on principal coordinates analysis (PCoA) of fecal samples from pigs after contact with seeder pigs (d0, n=15); 2 days pre-SD (d-2SD, n=15); 1 day pre-SD (d-1SD, n=15); and the day mucohaemorrhagic diarrhea was observed for the first time (MHD, n =14).** (R-squared: 0.091684; *p*-value: 0.003).

immunity in multiple sclerosis through IL-10 induction [54], and relieve intestinal inflammation in mice with acute and chronic colitis by reducing the levels of pro-inflammatory cytokines [39]. *Alistipes dispar* was also found depleted on d-1SD and d-2SD, when compared to MHD day. *Alistipes* spp. are enriched in human fecal samples from patients with colorectal cancer [55, 56], and other non-intestinal disorders, such as depression and atherosclerotic cardiovascular disease [57, 58]. It is suggested to thrive in the inflamed colon of IL-10$^{-/-}$ knocked out mice, being sufficient to induce colitis and tumorigenesis through IL-6–STAT3 signaling [59]. In contrast, decreased abundance of *Alistipes* spp. has been associated with protective effects in IBD patients and ulcerative colitis in mice [60, 61]. Interestingly, when *A. finegoldii* was administered together with *Bacteroides eggerthii*, a colitis-predisposing bacterium, it attenuated the severity of dextran sulfate sodium (DSS)-induced colitis in mice depleted of intestinal microbiota [60]. We speculate both anaerobes, *Parabacteroides* spp. and *Alistipes dispar*, increased abundance at MHD day is either linked to their opportunistic profile or they truly are part of the ancillary microbiota required for the severe expression of SD, as previously shown to be required [5–7, 9, 11].

Actinobacteria read counts decreased from d0 samples to MHD. This has not been reported before, although differences in analytical methods between this study and previous research

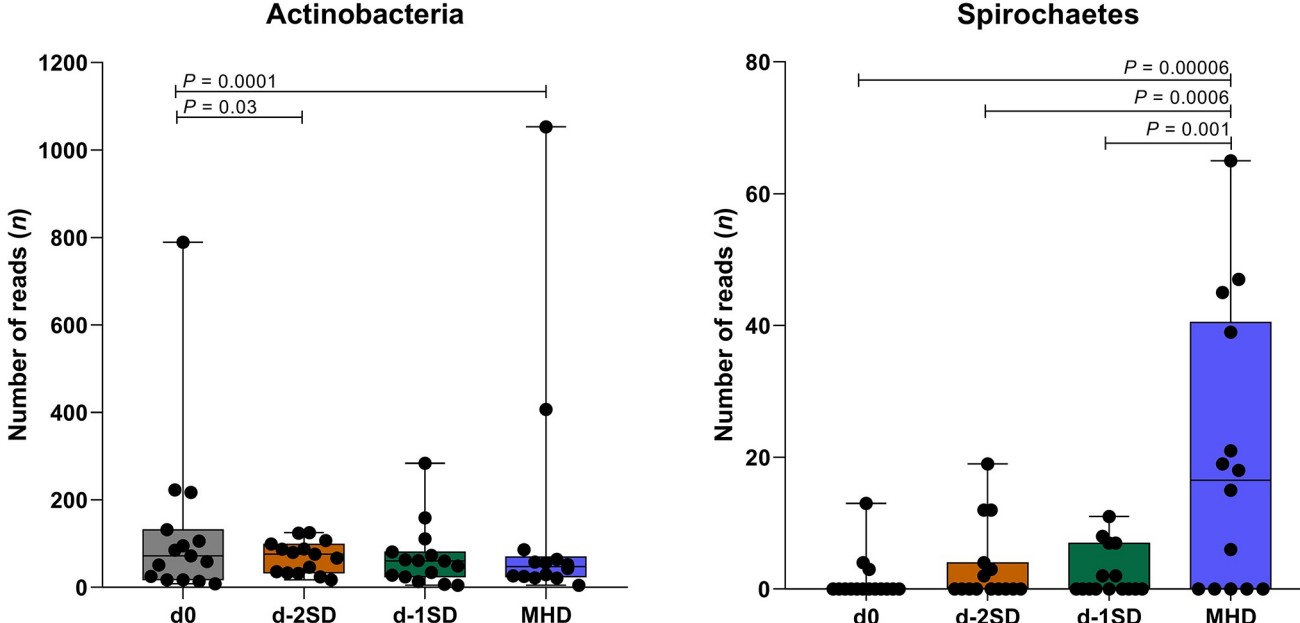

**Fig 4. Phyla significantly enriched or depleted in fecal samples after contact with seeder pigs (d0, n=14); 2 days pre-SD (d-2SD, n=15); 1 day pre-SD (d-1SD, n=15); and the day mucohaemorrhagic diarrhea was observed for the first time (MHD, n =14).** Boxplot represents the sum of reads associated with a given phylum at a given sampling time, and whiskers depict the minimum and maximum number of reads.

may explain such finding [11, 12]. Interestingly, a significant increase in Actinobacteria abundance and decreased incidence of clinical SD was observed in *B. hyodysenteriae*-inoculated pigs consuming a highly fermentable fiber, when compared to pigs fed a low fermentable fiber [21]. Several studies have found a decline in Actinobacteria abundance during different gastrointestinal disorders, such as acute hemorrhagic diarrhea in dogs [62], new neonatal porcine diarrhea [63], and post-weaning diarrhea in piglets [64]. Although this phylum has been found in higher abundance in healthy intestinal samples when compared to diseased ones, to clarify the exact role of actinobacteria in SD requires further studies.

Higher Chao1 index was identified on d-1SD, when compared to the control d0 samples. Differing from our findings, Burrough et al. [11] found a low Chao1 diversity index in colonic contents and mucosal scrapings of *B. hyodysenteriae* or *B. hampsonii* inoculated pigs, when compared to uninoculated controls. No changes in alpha diversity were observed in the fecal microbiota of pigs inoculated with *B. hampsonii* [12], *E. coli* F18+ [65, 66], or *S.* Typhimurium

**Table 1. Amplicon sequence variants (ASV) significantly different abundance between d-2SD and MHD.**

| ASV | Total read counts | | logFC[3] | adjusted *P*-value |
|---|---|---|---|---|
| | d-2SD[1] | MHD[2] | | |
| *Parabacteroides gordonii* | 6228 | 17377 | -1.524 | 0.002 |
| *Eubacterium brachy* | 7112 | 1546 | 2.141 | 0.002 |
| *Prevotella buccae* | 1967 | 6224 | -2.283 | 0.018 |
| *Alistipes dispar* | 1789 | 8254 | -3.373 | 0.002 |

[1]d-2SD: 2 days before mucohaemorrhagic diarrhea was observed (n=15)

[2]MHD: day mucohaemorrhagic diarrhea was observed for the first time (n=14)

[3]The degree of differential abundance is represented by $log_2$ fold change (logFC) between d-2SD and MHD samples.

**Table 2. Amplicon sequence variant (ASV) with significantly different abundance between d-1SD and MHD.**

| ASV | Total read counts | | logFC[3] | adjusted *P*-value |
|---|---|---|---|---|
| | d-1SD[1] | MHD[2] | | |
| *Alistipes dispar* | 3089 | 8254 | -3.652 | 0.002 |

[1]d-1SD: one day before mucohaemorrhagic diarrhea was observed (n=15)

[2]MHD: day mucohaemorrhagic diarrhea was observed for the first time (n=14)

[3]The degree of differential abundance is represented by $\log_2$ fold change (logFC) between d-1SD and MHD samples.

[67], when compared to matching controls. When compared to other alpha diversity indices, Chao1 index is considered sensitive to rare taxa [68, 69]. Given the inherent limitations of high-throughput sequencing, changes in ASVs with low abundance could be a simple result of the technique used to generate the data, or the bioinformatic algorithms used.

Beta-diversity analysis at the genus level revealed a single cluster with all samples intertwined. A previous study found no significant differences in beta-diversity fecal samples of pigs prior to inoculation and at the onset of mucohaemorrhagic diarrhea [12]. However, differences in beta-diversity between the luminal content and mucosal scrapings were observed in pigs with and without SD [11]. It is known that the mucosal, luminal and fecal microbiomes are compositionally different [70], which may have limited our findings. As a limitation of our study, the design used relied on *ante-mortem* samples. Unless surgical intervention was performed, which could confound the development of clinical signs as well as the gut microbiome, sampling the luminal or mucosal microbiome of pigs on the days prior to SD is rather challenging.

While we recognize that a small sample size was used in this study, our investigation revealed that the fecal microbiota changed in the days prior to the development of clinical SD. We suggest that the anaerobes *A. dispar* and *P. gordonii* may play a role in contributing to the development of SD. This knowledge may be employed for the future development of preventative tools that may not target the agents of SD, but still prevent clinical disease. Further investigation on their specific role may help clarify the importance of other microbes in SD. In addition, studies associating the fecal metabolome to the microbiota taxonomic composition may shed a light on the microbiota role in SD pathogenesis.

## Supporting information

**S1 Table. Description of swine dysentery custom diet.**
(PDF)

**S2 Table. Description of fecal samples used in this study.**
(PDF)

**S3 Table. Amplicon sequence variants (ASV) with significant differential abundance between d0 and all other days.**
(PDF)

## Acknowledgments

The authors are grateful to Champika Fernando and Scott do Santos for technical assistance and helpful discussions on data analysis.

## Author Contributions

**Conceptualization:** Matheus de O. Costa.

**Data curation:** Jéssica A. Barbosa, Matheus de O. Costa.

**Formal analysis:** Matheus de O. Costa.

**Funding acquisition:** Vinícius S. Cantarelli, Matheus de O. Costa.

**Investigation:** Jéssica A. Barbosa, Juan C. P. Aguirre, Roman Nosach, Matheus de O. Costa.

**Methodology:** Jéssica A. Barbosa, Juan C. P. Aguirre, Roman Nosach, John C. S. Harding.

**Project administration:** Matheus de O. Costa.

**Resources:** Matheus de O. Costa.

**Software:** Matheus de O. Costa.

**Supervision:** Matheus de O. Costa.

**Validation:** Jéssica A. Barbosa, Matheus de O. Costa.

**Visualization:** Jéssica A. Barbosa, Matheus de O. Costa.

**Writing – original draft:** Jéssica A. Barbosa, Matheus de O. Costa.

**Writing – review & editing:** Vinícius S. Cantarelli, Matheus de O. Costa.

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
