## [Decision Letter · Decision Letter 0]

15 Aug 2023

PONE-D-23-20538

Characterization of the bacterial fecal microbiota composition of pigs preceding the clinical signs of swine dysentery

PLOS ONE

Dear Dr. Costa,

Thank you for submitting your manuscript to PLOS ONE. After careful consideration, we feel that it has merit but does not fully meet PLOS ONE’s publication criteria as it currently stands. Therefore, we invite you to submit a revised version of the manuscript that addresses the points raised during the review process.

We look forward to receiving your revised manuscript.

Kind regards,

Gianmarco Ferrara, PhD, MVD

Academic Editor

PLOS ONE

Journal Requirements:

4. We notice that your supplementary figures are uploaded with the file type 'Figure'. Please amend the file type to 'Supporting Information'. Please ensure that each Supporting Information file has a legend listed in the manuscript after the references list.

Additional Editor Comments:

In particular, authors should focus on satisfying the comments of reviewer 1. 

Reviewers' comments:

Reviewer's Responses to Questions

**Comments to the Author**

1. Is the manuscript technically sound, and do the data support the conclusions?

Reviewer #1: Partly

Reviewer #2: Yes

Reviewer #3: Yes

2. Has the statistical analysis been performed appropriately and rigorously? 

Reviewer #1: No

Reviewer #2: Yes

Reviewer #3: Yes

3. Have the authors made all data underlying the findings in their manuscript fully available?

Reviewer #1: Yes

Reviewer #2: Yes

Reviewer #3: Yes

4. Is the manuscript presented in an intelligible fashion and written in standard English?

Reviewer #1: No

Reviewer #2: Yes

Reviewer #3: Yes

5. Review Comments to the Author

Reviewer #1: The manuscript is designed to characterize the intestinal microbiota of piglets prior to inoculum with the bacterium that causes swine dysentery. , and contrast the findings with the profiles of the microbiota of pigs exposed 1 or 2 days before the presence of hemorrhagic diarrhea .

I want to understand that the purpose is to define if there is a bacterial profile in these pigs that is permissive to develop hemorrhagic dysenteria. However, the purpose is not explicitly clear because the methodology does not mention a study design as such, on the other hand , although the hypothesis of the authors is mentioned .

The methodology section is mixed with what would be part of the results and in the section where the description of the results should be, the figures are described as the figure captions were.

Thus, the manuscript does not conform to the standards of the journal.

Reading the introduction section, I was struck by the fact that the authors mention that there is evidence on this same model, with results comparable to those presented in this manuscript.

In my opinion, the manuscript should be structured in a consistent way, making it clear: 1- what is the purpose and the novelty of this research strategy with respect to what has already been published; 2- the methodology section describe the methodology or, where appropriate, 3- gather methodology and results in a single section and 4- discuss the results and contrast them with those obtained by other authors, mention the limitations of the model and highlight the most important findings in a conclusion section

Reviewer #2: General comments:

This is very well written manuscript that has a clear study design. The main rationale for the study was to understand the change in the fecal microbiota few days before the development of mucohemorrhagic diarrhea due to swine dysentery in pigs. For the study, they used a seeder exposure method that it is interesting to mimic natural infection. Despite the difference in the days until the animals started with mucohemorrhagic diarrhea, fecal samples were collected daily, demonstrating the complexity of the study design. Chao1 Index between d-1SD and MHD compared to D0 was demonstrated as the only change in alpha diversity. Alistipes dispar and Parabacterioides gordonii increased in MHD sound as an interesting finding that may help to elucidate B. hyodysenteriae pathogenesis. It would be interesting to see some discussion about why P. gordonii has been observed in health pigs but had demonstrated a 3-fold increase in diseased animals (MHD).

Overall, this is an interesting study that will improve the knowledge about modifications observed in swine dysentery diseased pigs.

Specific comments:

Line 88: As diet is considered an important variable to the development of swine dysentery, it will be interesting to add supplementary table with the diet composition including ingredients and protein, energy, macro and microelements.

Reference 31 is not cited in the text. Please include this citation or delete it from the list of references.

Line 244: SCFAs is cited for the first time as an abbreviation. What does it mean?

Line 272: “….findings, Burrough et al. (11) found…”

Reviewer #3: The manuscript deals with a relevant and interesting theme. This topic will be of interest to the readership of PLOSONE.

Generally, the structure of each section of the paper is very clear and well defined.

The methods used are sound and properly described.

The results are interesting and well discussed.

The limits of the study are reported within the text with a sufficient discussion.

The English is correct and fluent for the readers.

Not all the images included in the paper are sharp enough. Authors should submit better quality figures.

I did not find the captions of figures, authors should provide them.

Lines 263 and 266: please modify “actinobacteria” in “Actinobacteria”

Line 287: ante-mortem should be italicized

6. PLOS authors have the option to publish the peer review history of their article (what does this mean?). If published, this will include your full peer review and any attached files.

Reviewer #1: **Yes: **CECILIA XIMENEZ MD PhD

Reviewer #2: No

Reviewer #3: **Yes: **Francesca Romana Massacci

---

## [Author Response · Author response to Decision Letter 0]

5 Oct 2023

We appreciate the reviewer’s unpaid time and efforts in reviewing this manuscript. Thank you!

Reviewer #1: The manuscript is designed to characterize the intestinal microbiota of piglets prior to inoculum with the bacterium that causes swine dysentery. , and contrast the findings with the profiles of the microbiota of pigs exposed 1 or 2 days before the presence of hemorrhagic diarrhea.

1- I want to understand that the purpose is to define if there is a bacterial profile in these pigs that is permissive to develop hemorrhagic dysenteria. However, the purpose is not explicitly clear because the methodology does not mention a study design as such, on the other hand , although the hypothesis of the authors is mentioned.

A: The purpose of the study is to characterize changes in the fecal microbiome immediately prior to disease. Previous studies have only characterized the changes in the microbiome of clinically affected pigs versus healthy controls or samples collected prior to inoculation (See change in L54). As stated in L74 (where a hypothesis and goal are described), we aimed to profile the changes in fecal microbiome composition immediately prior to the development of SD. 

2- The methodology section is mixed with what would be part of the results and in the section where the description of the results should be, the figures are described as the figure captions were. Thus, the manuscript does not conform to the standards of the journal. 

A: It is unclear to us as what the reviewer perceived as “methodology section is mixed with results”. Unfortunately, the Reviewer did not provide specifics, so we are unable to address this issue. That being said, we believe that they might refer to Table 1, which is truly part of the methods rather than results. Sampling (and the “result” from inoculation) is a key aspect of the methodology used in this study since we required a prior knowledge of clinical outcomes in order to analyze the correct samples (1 and 2 days prior to SD). 

PLOS One guidelines to authors states that “Each figure caption should appear directly after the paragraph in which they are first cited.”. The Reviewer can familiarize themselves with these guidelines by clicking here: https://journals.plos.org/plosone/s/submission-guidelines

3- Reading the introduction section, I was struck by the fact that the authors mention that there is evidence on this same model, with results comparable to those presented in this manuscript. In my opinion, the manuscript should be structured in a consistent way, making it clear: 1- what is the purpose and the novelty of this research strategy with respect to what has already been published; 

A: We regret that the introduction was not clearly written, leading to some confusion by the reviewer. We have modified the text (L54 and L73) to hopefully clarify it. Our results differ from other published work from the perspective that those studies did NOT sample pigs immediately prior to the development of disease. Pigs were sampled prior to inoculation, and not naturally infect but inoculated directly with an overload of pathogen. In our case, we analyzed samples from pigs that were naturally infected with Brachyspira, but were still in the incubation period (text modified in L77).

2- the methodology section describe the methodology or, where appropriate, 3- gather methodology and results in a single section. 

A: We hope to have clarified this in the comments above. 

4- discuss the results and contrast them with those obtained by other authors, mention the limitations of the model and highlight the most important findings in a conclusion section.

A: Thank you, this can be found in L223-300.

Reviewer #2: General comments:

This is very well written manuscript that has a clear study design. The main rationale for the study was to understand the change in the fecal microbiota few days before the development of mucohemorrhagic diarrhea due to swine dysentery in pigs. For the study, they used a seeder exposure method that it is interesting to mimic natural infection. Despite the difference in the days until the animals started with mucohemorrhagic diarrhea, fecal samples were collected daily, demonstrating the complexity of the study design. Chao1 Index between d-1SD and MHD compared to D0 was demonstrated as the only change in alpha diversity. 

Alistipes dispar and Parabacterioides gordonii increased in MHD sound as an interesting finding that may help to elucidate B. hyodysenteriae pathogenesis. It would be interesting to see some discussion about why P. gordonii has been observed in health pigs but had demonstrated a 3-fold increase in diseased animals (MHD).

A: Acknowledged. We have added more info on P. gordonii in L230-240. Unfortunately, there’s nothing very clear regarding how this bacterium could be contributing to the disease, or if it is simply an opportunistic.

Overall, this is an interesting study that will improve the knowledge about modifications observed in swine dysentery diseased pigs.

Specific comments:

Line 88: As diet is considered an important variable to the development of swine dysentery, it will be interesting to add supplementary table with the diet composition including ingredients and protein, energy, macro and microelements.

A: We have added the reference for the diet used in line 89 and the diet composition in a supplementary table 1 (S1_File). 

Reference 31 is not cited in the text. Please include this citation or delete it from the list of references.

A: Acknowledged. Reference 31 (updated for 32) included (Line 151). 

Line 244: SCFAs is cited for the first time as an abbreviation. What does it mean?

A: Thank you, we have addressed this in lines 254 and 255.

Line 272: “….findings, Burrough et al. (11) found…”

A: Acknowledged. Line 283. 

Reviewer #3: The manuscript deals with a relevant and interesting theme. This topic will be of interest to the readership of PLOSONE.

Generally, the structure of each section of the paper is very clear and well defined.

The methods used are sound and properly described.

The results are interesting and well discussed.

The limits of the study are reported within the text with a sufficient discussion.

The English is correct and fluent for the readers.

Not all the images included in the paper are sharp enough. Authors should submit better quality figures.

A: Acknowledged. Figures have been updated to publication quality.

I did not find the captions of figures, authors should provide them.

A: PLOS formatting guidelines is different from most journals. Captions are included in the text as figures are cited, instead of at the end of the manuscript. Figure 1 is in L167, Figure 2 L180, Figure 3 L186, and Figure 4 in L196.

Lines 263 and 266: please modify “actinobacteria” in “Actinobacteria”

A: Acknowledged (L274 and L277).

Line 287: ante-mortem should be italicized

A: Acknowledged (L298).

---

## [Editor Report · Decision Letter 1]

18 Oct 2023

PONE-D-23-20538R1Characterization of the bacterial fecal microbiota composition of pigs preceding the clinical signs of swine dysenteryPLOS ONE

Dear Dr. Costa,

Thank you for submitting your manuscript to PLOS ONE. After careful consideration, we feel that it has merit but does not fully meet PLOS ONE’s publication criteria as it currently stands. Therefore, we invite you to submit a revised version of the manuscript that addresses the points raised during the review process.

We look forward to receiving your revised manuscript.

Kind regards,

Gianmarco Ferrara, PhD, MVD

Academic Editor

PLOS ONE

Journal Requirements:

Additional Editor Comments:

The authors addressed the reviewers' comments satisfactorily. Below are some of my minor comments preceding acceptance.

Abstract:

I’m not sure that readers can understand what the authors intend for “seeder pig” and “contact pigs” reading only abstract section.

The abstract can’t finish with this statement “Future investigations to verify the specific role of these taxa on the pathogenesis of SD is warranted” but requires the implications and the limitations of this study in details.

Introduction:

Line 40: Delete “initial”

Lines 43-49: This list of studies should be preceded by an introductory sentence such as "Numerous studies in the literature have demonstrated the role etc."

Line 56: italics?

Line 63: I advise the authors to reverse "colonic microbiome" and "incidence of SD".

Line 73-75: Change this sentence as follows: We assessed any disturbances in the fecal microbiome immediately before clinical SD was observed.

Authors should specify what they consider “immediately prior to the clinical SD”. Was it during incubation? How many days?

Discussion:

Line 225: Please change “stepping-stone” in something else.

What are the implications of this study? Understanding the microbiome in order to intervene in some way?

The limitation of the few samples analyzed was discussed, not the fact that many parameters would not seem to differ between infected and non-infected animals.

---

## [Author Response · Author response to Decision Letter 1]

25 Oct 2023

The authors addressed the reviewers' comments satisfactorily. Below are some of my minor comments preceding acceptance.

Abstract:

I’m not sure that readers can understand what the authors intend for “seeder pig” and “contact pigs” reading only abstract section.

A: Addressed.

The abstract can’t finish with this statement “Future investigations to verify the specific role of these taxa on the pathogenesis of SD is warranted” but requires the implications and the limitations of this study in details.

A: Updated.

Introduction:

Line 40: Delete “initial”

A: Deleted.

Lines 43-49: This list of studies should be preceded by an introductory sentence such as "Numerous studies in the literature have demonstrated the role etc."

A: Updated.

Line 56: italics?

A: Phyla names do not need to be itacilized.

Line 63: I advise the authors to reverse "colonic microbiome" and "incidence of SD".

A: Updated.

Line 73-75: Change this sentence as follows: We assessed any disturbances in the fecal microbiome immediately before clinical SD was observed.

Authors should specify what they consider “immediately prior to the clinical SD”. Was it during incubation? How many days?

A: Addressed (L75).

Discussion:

Line 225: Please change “stepping-stone” in something else.

A: Changed to “intermediary step”

What are the implications of this study? Understanding the microbiome in order to intervene in some way?

The limitation of the few samples analyzed was discussed, not the fact that many parameters would not seem to differ between infected and non-infected animals.

A: We refrained from overinterpreting the data. Microbiome-host-pathogen interactions are extremely complex and cannot be fully dissected with a single study looking at the bacteriome. We have addressed this issue in lines 306-307. 

It is unclear what the editor means by “many parameters”? If this is related to alpha/beta diversity, that is expected given the short amount of time between sample collection points. All studies published to date looking at microbiome changes due to SD sampled animals at peak clinical signs vs before inoculation, when lesions and disease are well underway (which takes 7-14 days). In the case of our study, the very short period between d-2 and clinical disease (MHD, 48h) likely contributed to the lack of differences between parameters – which was expected. In addition, the editor may want to revise the methodology employed here: there is no infected vs non-infected comparison in this study.

---

## [Editor Report · Decision Letter 2]

30 Oct 2023

Characterization of the bacterial fecal microbiota composition of pigs preceding the clinical signs of swine dysentery

PONE-D-23-20538R2

Dear Dr. Matheus O. Costa,

We’re pleased to inform you that your manuscript has been judged scientifically suitable for publication and will be formally accepted for publication once it meets all outstanding technical requirements.

Kind regards,

Gianmarco Ferrara, PhD, MVD

Academic Editor

PLOS ONE

---

## [Editor Report · Acceptance letter]

2 Nov 2023

PONE-D-23-20538R2 

Characterization of the bacterial fecal microbiota composition of pigs preceding the clinical signs of swine dysentery 

Dear Dr. Costa:

I'm pleased to inform you that your manuscript has been deemed suitable for publication in PLOS ONE. Congratulations! Your manuscript is now with our production department. 

Kind regards, 

on behalf of

Dr. Gianmarco Ferrara 

Academic Editor

PLOS ONE